# LEARNING TO PLAN WITH PERSONALIZED PREFERENCES

## ABSTRACT

Understanding and adapting to human preferences is essential for the effective integration of Artificial Intelligence (AI) agents into daily human life, particularly in collaborative and assistive roles. Previous studies on embodied intelligence have primarily adopted a generalized yet non-personalized approach. Our research aims to address this gap by focusing on endowing agents with few-shot learning capabilities for human preferences and generalizable **planning guided by learned preferences**, as individual preferences are often implicitly described in minimal observations, and abstract enough to generalize across situations. To study such formulation, we introduce Preference-based Planning (PbP), an embodied environment supporting hundreds of diverse preferences ranging from complex action sequences to specific sub-actions. By benchmarking State-of-the-Art (SOTA) methods on PbP, we demonstrate that while symbol-based approaches show promise in terms of effectiveness and scalability, few-shot learning of personalized preferences and planning with adaptive actions remain challenging. Our findings further reveal that incorporating preference as a key intermediate representation in planning can significantly improve the personalization and adaptability of AI agents. These results establish preference as a valuable abstraction of human behaviors and pave the way for future research on more efficient preference learning and personalized planning in dynamic environments.

## 1 INTRODUCTION

The field of embodied Artificial Intelligence (AI) is rapidly advancing, driven by significant progress in foundation models for vision and language (Bommasani et al., 2021; Peng et al., 2023; Achiam et al., 2023; Bai et al., 2023). These advancements enable AI systems to autonomously collaborate with or assist humans in daily tasks, particularly in domestic settings (Driess et al., 2023; Leal et al., 2023; Zitkovich et al., 2023; Ahn et al., 2024). However, a critical aspect—personalization—remains inadequately addressed. Personalization is crucial for tailoring agent actions to individual users' unique preferences and needs, thereby significantly enhancing user satisfaction (Lee et al., 2012; Leyzberg et al., 2014).

The concept of "preference" is fundamental to personalization (Slovic, 1995), guiding human-like decision-making and intelligent behavior. Psychological research has shown that understanding preferences is crucial for interpreting and predicting human behaviors (Fawcett and Markson, 2010), as well as facilitating social cognition and interactions (Gerson et al., 2017; Liberman et al., 2021). For building embodied assistants, the ability to understand preferences could lead to deeper comprehension of human behavior and more grounded planning. However, recent attempts through natural language instructions (Mu et al., 2023; Zitkovich et al., 2023; Singh et al., 2023) may not suffice for capturing human preferences. While natural language is the most common method for humans to articulate their needs, its inherent ambiguity creates a gap between given instructions and actual executions. Embodied agents often require additional details to understand the instructor's intentions and act accordingly. For example, when a user requests assistance in preparing an apple to eat, the agent needs explicit information about apple selection (if there are multiple), washing requirements, cutting preferences, and container needs. These details, corresponding to users' preferences, vary from person to person. See Figure 1 for an example.

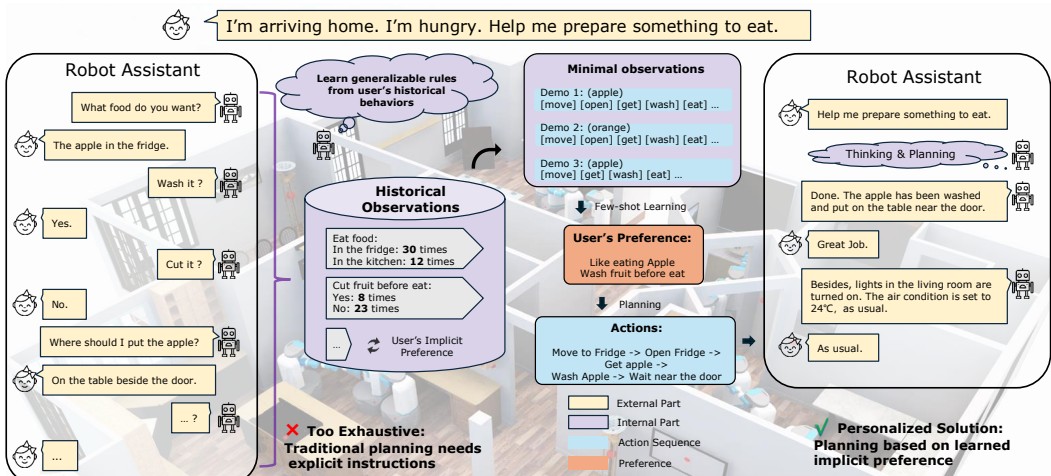

Figure 1: **An example of preference-based planning.** An agent is assigned the high-level task of "help prepare food." Traditional assistant agents require detailed instructions to explicitly fulfill human needs, which can be exhausting. Our proposed framework leverages preferences to aid in planning. By learning preferences as a key intermediate representation from minimal human demonstrations, our approach enables AI agents to deliver personalized and adaptable assistance.

Besides, accurately capturing and learning human preferences in real-world settings is challenging. Humans often communicate their needs succinctly, without exhaustive details about their preferences (Lichtenstein and Slovic, 2006), and their preferences may include unconscious or instinctive elements that are difficult to articulate fully (Epstein, 1994; Simonson, 2008). A more practical idea is to infer preferences from human choices and decision-making tendencies. In Figure 1, the robot assistant can infer the users' preference for apples and washing the fruit before eating from historical observations.

In pursuit of an intelligent and personalized embodied agent, we concentrating on the capabilities of first learning preferences from human behavior and subsequent planning guided by these learned preferences. We propose that integrating preferences into planning will improve efficiency and user satisfaction. While this problem is not completely untouched, previous studies such as NeatNet (Kapelyukh and Johns, 2022) and SAND (Yuan et al., 2023) are limited to a single task like rearrangement and fail to generalize across different situations. The gap between the limited study and the need for personalized agents motivates us to develop a comprehensive environment for embodied agents to learn human preferences that can be applied to various everyday tasks.

Therefore, we first introduce Preference-based Planning (PbP), a realistic embodied environment built upon NVIDIA Omniverse and OmniGibson simulation environment (Li et al., 2023). With the Omniverse's support, our PbP provides realistic simulation and real-time rendering for thousands of daily activities in 50 different scenes, making it an ideal foundation for building preference-based learning agents. Of the thousands of activities, we reference Behavior-1K and build a parameterized preference vocabulary of 290 diverse preferences. In this combinatorial space of preference, we identified hierarchical preferences, ranging from the action level to the task sequence level. Action-level preferences focus on specific items or attributes, such as the user's preferred type of glass or the desired water temperature, whereas task sequence-level preferences pertain to the user's preferred order of task execution or the prioritization of certain sub-tasks over others.

With the PbP developed, we challenge existing learning agents on their ability to learn human preference and subsequently conduct preference-based planning. Given the expensive data collection(Akgun et al., 2012) and the few-shot nature of imitation and induction in acquiring human preference, we frame the preference learning task as few-shot learning from demonstration. In this framework, an agent must adeptly respond to ambiguous task instructions and formulate adaptive task planning aligned with user preferences demonstrated in a few example action sequences. Ideally, the agent should analyze data from observing user behaviors, identify consistent patterns, and extrapolate from these behavioral consistencies to a higher-level abstraction of user preferences. Importantly, these preferences should be generalizable across various tasks and not tied to specific situations(Chao

et al., 2011). Furthermore, when faced with a new task, the agent should generate an adaptive plan of action sequences based on its understanding of the user's preferences to complete the task.

In our evaluation of state-of-the-art algorithms within the realm of preference-based planning tasks using PbP, we have discovered that preferences serve as a valuable abstraction of human behaviors. Incorporating preferences as a key intermediary step in planning can significantly enhance the capability and adaptability of AI agents. However, there are still significant challenges that current AI systems face. These difficulties stem not only from the complexities inherent in planning activities but also from the intricate process of learning and abstracting human preferences through perception. These dual challenges highlight the gap between the current capabilities of AI in understanding nuanced human preferences and the sophisticated demands of these tasks. We hope that our work will serve as a foundational step towards addressing these challenges.

## 2 RELATED WORK

### 2.1 EMBODIED ASSISTANTS

Developing intelligent embodied assistants that are capable of interpreting natural language instructions and executing corresponding actions in physical environments has been a cornerstone of robotics research. This journey began with the exploration of Vision-and-Language Navigation (VLN) tasks, wherein robots are trained to navigate in environments based on natural language instructions (Anderson et al., 2018; Chen et al., 2019; Thomason et al., 2020). Further, the scope of embodied tasks is extended to more interactive abilities beyond navigation. ALFRED (Shridhar et al., 2020) involves interactions with objects, keeping track of state changes, and references to previous instructions. Habitat (Savva et al., 2019; Puig et al., 2023) and AI2-THOR (Kolve et al., 2017) have advanced the field by emphasizing active perception, the necessity of long-term strategic planning, and the acquisition of knowledge through interactive learning. In recent developments, the research focus has been gradually shifted towards housekeeping tasks where explicit instructions are often absent, thus asking for robots to engage in more complex reasoning processes, mainly regarding the rearrangement of objects (Kapelyukh and Johns, 2022; Kant et al., 2022; Sarch et al., 2022; Wu et al., 2023). Some recent works (Patel and Chernova, 2023; Patel et al., 2023) have also focused on robots' anticipating temporal patterns of object movements associated with humans' everyday routines.

### 2.2 PREFERENCE-BASED PLANNING IN EMBODIED TASKS

The notion of "preference" employed varies in scope and application. On one hand, existing research primarily explores **general** preference-based action or task-planning. For example, Deep Reinforcement Learning from Human Preferences (Christiano et al., 2017) utilizes overall human preferences to deduce optimal action sequences according to typical conventions. Among one of the most studied is rearranging objects, where robots rely on commonsense knowledge to organize objects in a manner that reflects objects' common occurrence and placement within an environment (Taniguchi et al., 2021; Sarch et al., 2022). In such contexts, the word "preference" advocates generic interpretation, that is, universally accepted behavior norms in humans. On the other hand, the concept of "preference" also encompasses **personalized inclinations**, emphasizing that embodied agents' actions are not only efficient but also aligned with the nuanced habits of specific users. For example, recent works (Abdo et al., 2015; Kapelyukh and Johns, 2022; Wu et al., 2023) study the rearrangement of objects based on individualized placement strategies. Our work falls into the second context but also extends the task into more diverse situations and environments, including not only the spatial arrangements of objects but also the temporal sequence of interactions, the state changes during interactions, and the formulation of few-shot learning from demonstration.

### 2.3 LLMS AND VLMS IN EMBODIED TASKS

Large language models (LLMs) that are trained on massive text data (Shanahan, 2024), exhibit strong capacities to understand natural language and solve tasks through text generation (Zhao et al., 2023). Many recent works (Song et al., 2023; Driess et al., 2023; Ding et al., 2023) have explored using LLMs as few-shot planners to generate language plans for embodied tasks given a few demonstrations. Following the advances in natural language processing, vision-language models (VLMs) pre-trained

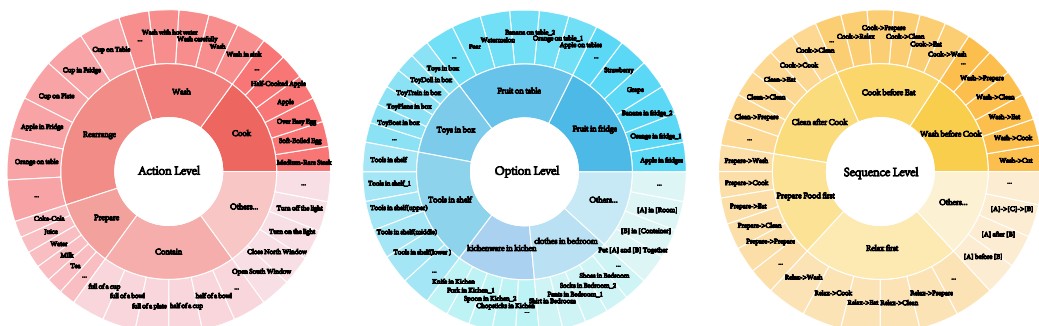

Figure 2: **Preferences and their distributions at different levels.** Zoom in to see more details.

with large-scale image-text pairs have appeared, and they can be directly applied to downstream visual tasks (Zhang et al., 2024). Recent methods integrating foundation VLMs towards embodied scenarios have also significantly improved a robotic system's perception and reasoning ability, enabling them to assist humans in everyday tasks (Ahn et al., 2024; Leal et al., 2023; Gu et al., 2023; Brohan et al., 2022; Zitkovich et al., 2023). However, while these foundation models are equipped with much common knowledge to reason from either text or image information, whether they can learn human preferences from a few examples remains to be tested.

## 3 THE PREFERENCE-BASED PLANNING (PBP) ENVIRONMENT AND DATASET

PbP inherits NVIDIA's Omniverse and OmniGibson simulation environment (Li et al., 2023) and supports realistic simulation and real-time rendering for thousands of daily activities in 50 different scenes. In the following, we detail how the preferences in the environment are defined and how the evaluation dataset is constructed.

### 3.1 DEFINITION OF PREFERENCES

We define preferences on a three-tiered hierarchical structure, covering various degrees of specificity and relevance across tasks. See Figure 2 for an overview of all defined preferences and their distribution. Figure 3 shows corresponding preferences and the agent's actions in the environment.

**Action Level** The bottom level preferences pertain to fine-grained actions in a sub-task. It deals with details in the process of a specific sub-task, such as the desired amount of water to fill in a cup or which level of a bookshelf to put a book on.

**Option Level** The middle level of preferences contain alternatives for a specific sub-task. Take rearrangement as an example. For "storing-nonperishable-food", some people prefer to put them in cabinets, while others may favor stacking them on the kitchen table. Note that preferences defined at this level could be bound to different objects and may be composed of multiple action-level preferences.

**Sequence Level** The top level of the hierarchy concerns the preference over sub-task order or prioritization of certain sub-tasks over others in one task. It encapsulates users' preferences about which sub-tasks should be undertaken first and which sub-tasks later. For example, the preference to clean the furniture first, then rearrange kitchen utensils, and finally prepare dinner, after returning home.

### 3.2 CONSTRUCTING PBP TASKS

Instead of recruiting human subjects, with the preferences defined and the environment ready, we sample from the preference primitives above and construct PbP tasks. We choose the Fetch robot with an articulated arm for grasping items and manipulating objects as our embodied agent in the

Figure 3: **Example of preferences and their corresponding actions in PbP.** Cooking using microwave (a), washing in the sink (b), and cutting to halves (c) belong to the primitive action level preferences. For rearranging objects, we present two option-level preferences of either grouping objects by their categories (d) or putting them on the same layer of the fridge (e). For after-dinner activities, a user might want to first have fruits and do some cleaning in the order shown in the sequence-level preference (f).

simulator. Specifically, to emulate the few-shot nature of preference-based planning in the real world, for each sampled task, we pair it with a few demonstrations that share the high-level preference but not exactly the same trajectory in terms of the objects selected or the scene used. When constructing a task from a sampled preference, we randomly assign it to one of the 50 different scenes provided by OmniGibson, and subsequently sample the objects bound to the preference. Once the scene, objects, and the preference are established, we generate egocentric observation and actions sequences of our embodied agent. When generating the demonstrations, the agent is guided by a manually designed rule-based planner. A set of planning primitives are used to simplify the process and focus on high-level planning, *e.g.*, Inverse Kinematics (IK) is employed for grasping, while the A* algorithm for movement. See Appendix B.2 for more implementation details.

Paired with the egocentric video of an agent's activity is its bird's-eye-view map of the position of the agent and frame-level textual annotation of the current action, as shown in Figure 4. Additionally, we provide a rendered third-person view of the entire process for better illustration. We choose the egocentric view of the agent in the simulator as the main input in the dataset for two primary reasons: 1) the egocentric perspective provides a clear view with minimal occlusions, and 2) it mimics a human's view, making the dataset and models easily transferable to real-world data collected by head-worn devices.

In total, PbP comprises 290 unique preferences categorized into three distinctive levels as detailed in Section 3.1. Among them, 80 are from the sequence level, 135 are from the option level, and 75 are from the action level. PbP comprises 15,000 unique egocentric instances(videos) as demonstrations of preferences for inference and learning.

## 4 FORMULATING PREFERENCE-BASED PLANNING

A PbP task resembles a real-world watch-and-help setting, where an agent is presented a few demonstrations of a user performing the same task and then asked to accomplish it in a different setup but following the preference of the user implied in the demonstrations.

### 4.1 TASK FORMULATION

Preference-based planning comprises two parts: few-shot **preference learning** of the user's preference and subsequent **planning** in a different environment based on the learned high-level preference. Humans, even infants are found to have the ability to detect others' preferences from their decisions (Choi and Luo, 2023). And it is nearly impossible to collect a large amount of demonstrations for a

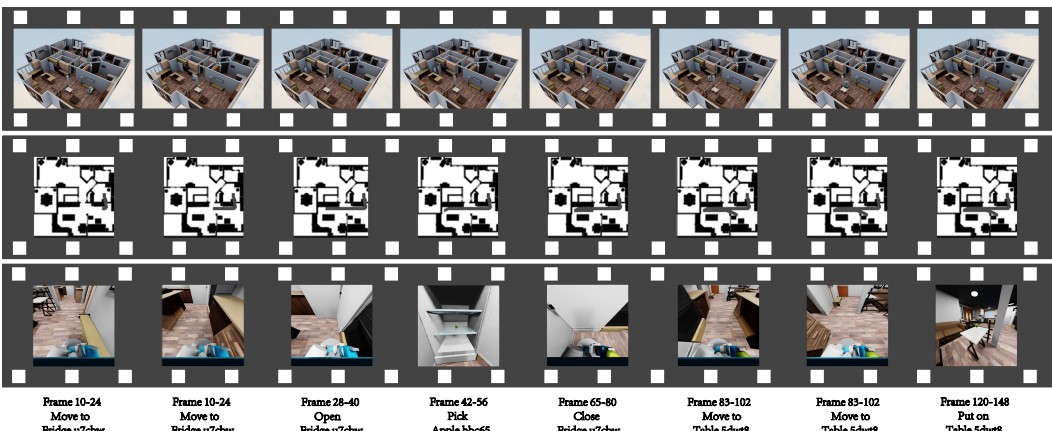

| Frame 10-24 | Frame 10-24 | Frame 28-40 | Frame 42-56 | Frame 65-80 | Frame 83-102 | Frame 83-102 | Frame 120-148 |
| Move to | Move to | Open | Pick | Close | Move to | Move to | Put on |
| Fridge u7cbw | Fridge u7cbw | Fridge u7cbw | Apple hbc65 | Fridge u7cbw | Table 5dwt8 | Table 5dwt8 | Table 5dwt8 |

Figure 4: **Example of a task in PbP.** The task is "Pick Apple from Fridge and place on Table". **Top:** The third-person view video looking down at the entire scene. **Middle:** The bird's-eye-view map showing the relative position of the robot in the scene. **Bottom:** The egocentric video of the robot's observation. **Text:** Extra per-frame textual annotation of actions.

specific person and task in daily life. We therefore formulate the problem as few-shot learning from demonstration. Given a certain user whose preference is denoted as $\mathbf{P}$, the agent observes the user performing a task in the first-person view, denoted as $\mathbf{O}$. The user may perform the task a few times. Formally, the input observation $\mathbf{O}$ contains both state and action observation, $\{(\mathcal{S}_i, \mathcal{A}_i, \mathcal{M})_N\}$ where $\mathcal{S}_i$ denotes the egocentric observation sequence in the $i$-th demonstration, $\mathcal{A}_i$ the action sequence, and optionally auxiliary bird-eye-view of the environment map $\mathcal{M}$.

Ideally, the objective in the first stage is to learn the preference representation demonstrated in the user actions, *i.e.*,

$$\mathbf{p} = f(\mathbf{O}; \theta_f), \tag{1}$$

where $\mathbf{p}$ denotes the preference representation.

The learned preference $\mathbf{p}$ should guide planning when the agent is placed in a setup with either different objects, room layouts, or even the entire environment. Specifically, we expect the agent to optimize

$$\mathcal{L} = \sum_{i=1} \ell(g(s_i, f(\mathbf{O}; \theta_f); \theta_g), a_i), \tag{2}$$

where $g(\cdot)$ denotes a potentially parameterized planning function that takes the current state and the preference representation and predicts the next action. $a_i$ denotes the ground-truth action demonstrating the user's preference at the current stage.

In the experiments, we assess models in both end-to-end and two-stage learning-planning settings to evaluate their performance in PbP. In the end-to-end setting, models learn to directly map raw state input to action output. Following the observation of models being able to perform in-context learning, we directly supply the demonstrations together with the current state as the input and optimize the cross-entropy loss of this output with the ground-truth result. Conversely, in the two-stage setting, models are provided with explicit preference labels during training and are trained to first separately predict what preference is shown in the examples; the predicted labels will then be used as the preference representation for predicting the action. For black-box models, we design prompts rather than fine-tuning.

## 4.2 MODELS

In this work, we primarily focus on multimodal models with a large language model component and proven tracks of capabilities in few-shot learning. Presumably, the language model part serves as the knowledge base and could boost preference learning from commonsense scenarios. Additionally, we include symbol-based Large Language Model (LLM) models for ablative purposes, allowing us to examine the impact of different modalities on PbP. Note that most models considered could be used in both the end-to-end and the two-stage pipeline.

**ViViT** We select the pure-Transformer-based Video Vision Transformer (ViViT) (Arnab et al., 2021) as a vanilla end-to-end trainable model, whose ability to extract spatial and temporal information on video input has been validated in a variety of tasks. Without a language model component, the model could be potentially inferior than others in understanding commonsense and hence should serve as the lower bound in PbP.

**LLaVA** Large Language and Vision Assistant (LLaVA) (Liu et al., 2024) is an end-to-end trainable large multimodal model that combines vision and text input for general-purpose visual and language understanding. The variant has significantly improved zero-shot reasoning capability with multimodal input. We test LLaVA-NeXT which has been finetuned for zero-shot video understanding.

**EILEV** Efficient In-context Learning on Egocentric Videos (EILEV) (Yu et al., 2023)'s in-context learning for egocentric videos emerges via architectural modification of a pretrained Vision-Language Model (VLM). We use a pretrained EILEV model with OPT-2.7B (Zhang et al., 2022) as the language backbone. The model is pretrained on Ego4D (Grauman et al., 2022), matching the egocentric view of the input video from PbP.

**GPT-4V** We also test GPT-4V model which has proven to be a strong reasoner in understanding image and video content. We run the model via the Azure OpenAI API with the GPT version "gpt-4-turbo-2024-04-09". Due to the image token limit, we subsample the input videos.

The following models are single-modal using the action sequences only.

**DAG-Opt** For symbolic reasoning based on the text input, we view the problem as a DAG-Optimization (denoted as DAG-Opt) task to learn the structure of dependency relations behind the actions (Zheng et al., 2018). We use a score-based NOTEARS model to learn a generalized Structural Equation Model (SEM) to help reasoning. We follow the few-shot setting in ACRE (Zhang et al., 2021) to perform reasoning based on the learned causal dependency structure.

**LLMs** We also evaluate modern language-based foundation models such as Llama3 (Touvron et al., 2023) and GPT-4 (Achiam et al., 2023), leveraging the pure action input from PbP. The action input serves as a high-level abstraction of the egocentric video, reducing the complexity associated with visual data. In particular, we utilize Llama3-8B as the baseline and GPT-4-Turbo as the state-of-the-art for comparitive purposes. The prompt design is mainly motivated by OpenAI Cookbook [1]. See Appendix D for more details about the model structures and the prompt design.

## 5 EXPERIMENTS

### 5.1 EXPERIMENTAL SETUP

As discussed in Section 4.1, we evaluate models' few-shot preference learning in two different settings. In the end-to-end setting, models are required to generate the action sequence directly from the historical observations. In the two-stage learning-planning setting, models are tasked with predicting the preference label first and then use the predicted preference to plan actions. We provide all models with three video demonstrations illustrating a specific preference for few-shot learning. All videos are egocentric with $512 \times 512$ resolutions. All language models decode with temperature of 0.05, top-k of 1, and top-p of 0.05. Training and inference for all models are conducted on one machine with 8 NVIDIA A100 cards.

Table 1: **Levenshtein distance between the generated action sequences and the ground truth.**

| | | VIDEO-BASED INPUT | | | | SYMBOL-BASED INPUT | |
|---|---|---|---|---|---|---|---|
| | | ViViT | LLaVA-Next | EILEV | GPT-4V | Llama3-8B | GPT-4 |
| **Option Level** | **End-to-end** | 15.49±1.29 | 15.94±3.41 | 12.88±2.20 | 15.63±2.31 | 14.74±3.21 | 12.23±2.96 |
| | **Second-stage** | - | 3.28±5.29 | 11.18±4.20 | 1.26±2.55 | 8.22±5.58 | 0.12±3.12 |
| **Sequence Level** | **End-to-end** | 34.04±11.84 | 34.76±11.25 | 33.10±12.21 | 33.75±11.15 | 31.79±7.32 | 27.85±6.57 |
| | **Second-stage** | - | 18.92±14.18 | 26.57±12.21 | 11.36±8.05 | 19.02±7.10 | 12.29±3.12 |
| **Overall** | **End-to-end** | 24.76 | 25.35 | **22.99** | 24.69 | 23.26 | **20.04** |
| | **Second-stage** | - | 11.10 | 18.88 | **6.31** | 13.62 | **6.21** |

---

[1] https://github.com/openai/openai-cookbook.git

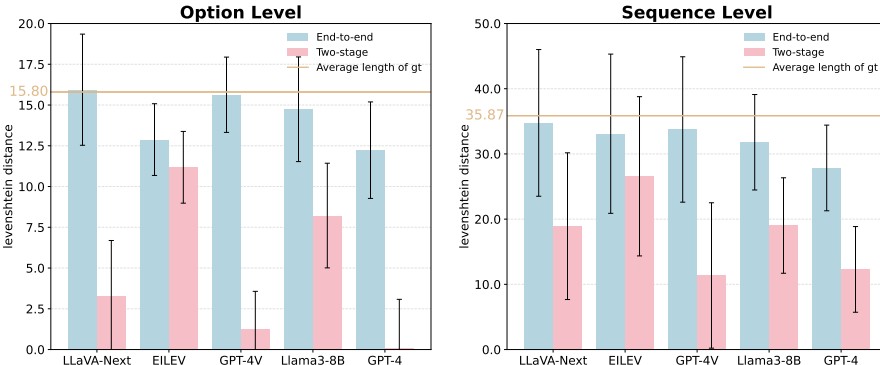

Figure 5: **Levenshtein distance between the generated action sequences and the ground truth.** Models are evaluated under two settings. In the end-to-end setting, models are required to generate the action sequence directly from the historical observation, while in the second stage of the two-stage setting, models are provided with the predicted preference labels.

## 5.2 END-TO-END ACTION PREFERENCE LEARNING

We first show model performance under the end-to-end setting. A model directly generates an action based on historical demonstrations and the current state in this setting. We employ the Levenshtein distance as the metric to measure the discrepancy between the ground truth and the generated action sequence, where we treat each single action as a token. As illustrated in Table 1 (the End-to-end line), all video-based models' results incur Levenshtein distances closer to the average sequence length (option level 15.80, sequence level 35.87), suggesting that they predominantly predict inconsistent actions and do not understand the preferences implied in the demonstration videos. Symbol-based models demonstrate relatively better performance, although the improvement is marginal. The experimental results here imply that existing models still fall short in inferring hidden relations from perceptual input without explicit intermediate outcome; they can only learn a few isolated actions rather than entire action patterns based on implicit preferences.

## 5.3 TWO-STAGE LEARNING-PLANNING

Table 2: **The accuracy of preference prediction.** All models are tested in the few-shot setting.

| | VIDEO-BASED INPUT | | | | SYMBOL-BASED INPUT | | |
|---|---|---|---|---|---|---|---|
| | ViViT | LLaVA-Next | EILEV | GPT-4V | DAG-Opt | Llama3-8B | GPT-4 |
| **Option Level** | 9.38 | 36.87 | 38.33 | 48.48 | 10.15 | 72.98 | 86.27 |
| **Sequence Level** | 4.24 | 24.85 | 32.69 | 37.50 | 13.49 | 67.18 | 68.42 |
| **Overall** | 6.81 | 30.86 | 35.51 | **42.99** | 11.82 | 70.08 | **77.34** |

Due to the limitations of direct end-to-end learning, we simplify the preference learning problem using a two-stage approach. In the first stage, we provide the preference prediction module with the auxiliary preference token labels, and explicitly train the preference learning module to correctly predict the hidden preferences. As to our preference hierarchy discussed in Section 3.1, the preference tokens are semantic enough to be translated into primitive actions. The experimental results are summarized in Table 2. As shown in the table, video-based models' performance largely varies across different levels. At the option level, GPT-4V outperforms other models with an accuracy of 48.48, demonstrating the strongest capability in deciphering the preference implied in demonstrations. For symbol-based models, the relatively poor performance of dependency-based DAG-Opt and the much improved performance in Llama3-8B and GPT-4 marked the significant difference between dependency learning and next-token prediction in inferring preferences. Despite the differences in the performance between the models, models with a language model component like LLaVA-Next and GPT-4 display better understanding of preference when compared to previous direct end-to-end learning.

In the second stage, a model generates action sequences based on both past demonstrations and the current preference label. The results of the Levenshtein distance are presented in Table 1 (the

Second-stage line), and for a more intuitive comparison, refer to Figure 5. Notably, at both the option level and the sequence level, most models show significant improvement in their planning abilities when provided with explicit preferences. In particular, GPT-4V and GPT-4 demonstrate almost zero distance between their predicted actions and the ground truth, indicating that most of their planning match the ground truth action sequence.

Combining the results of both stages and comparing them with end-to-end learning, we suggest potential reasons for the initial poor performance of different models. Vision-based models like LLaVA-Next and GPT-4V exhibit low accuracy in inferring preferences but show substantial improvement when generating action plans with preference labels. This observation indicates that they struggle to extract abstract preference information from raw visual observations. On the other hand, symbol-based models perform reasonably well in both preference inference and planning with preferences. However, they still fall short under an end-to-end setting. This finding implies that when personalized preferences are explicitly taken into account, current models' few-shot planning in tasks involving preferences may be effective, as they seem to lack this mode of thinking.

To understand how much of performance comes from the prior knowledge, especially for pretrained models, and how much is from the in-context demonstrations, we also conduct an ablation study. In particular, we remove all the demonstrations in the input and instead supply the model with a test sequence to see how well it can predict the preferences in the test sequence. The experimental outcome is presented in Table 3. Comparing Table 2 and Table 3, all models experience significant performance drop, suggesting that models do extract meaningful information for preference prediction. Notably, models undergo severe performance decline in the sequence level, which indicates that while optional choices for a specific task might have been encoded in prior knowledge in models, when there is more variability in sequences, models do have to use in-context examples to recognize the hidden relations.

Table 3: **The accuracy of preference prediction.** All models are tested in the ablative setting, where we remove the demonstrations and assess the models on test sequences only.

| | VIDEO-BASED INPUT | | | | SYMBOL-BASED INPUT | | |
|---|---|---|---|---|---|---|---|
| | ViViT | LLaVA-Next | EILEV | GPT-4V | DAG-Opt | Llama3-8B | GPT-4 |
| **Option Level** | 9.16 | 15.47 | 4.77 | 29.42 | 3.84 | 39.50 | 73.87 |
| **Sequence Level** | 4.38 | 8.13 | 0.00 | 0.00 | 1.28 | 6.25 | 9.42 |
| **Overall** | 6.77 | 11.8 | 2.38 | **14.71** | 2.56 | 22.88 | **41.64** |

## 5.4 GENERALIZATION

Table 4: **Models' generalization ability.** *direct* denotes experiments conducted *without* generalization. *orig* denotes the original experiments conducted *with* generalization cases. Also the accuracy of preference prediction.

| | LLaVA-Next | EILEV | GPT-4V | GPT-4 |
|---|---|---|---|---|
| **Option Level** *direct* | 33.25 | 46.93 | 53.24 | 86.32 |
| **Option Level** *orig* | 36.87 | 38.33 | 48.48 | 86.27 |
| **Sequence Level** *direct* | 33.12 | 37.53 | 39.42 | 70.27 |
| **Sequence Level** *orig* | 24.85 | 32.69 | 37.50 | 68.42 |

Human actions may vary with different objects in various scenes, yet their preferences could remain consistent. Therefore, we also investigate the models' generalization ability in learning human preferences within different scenes. It's worth noting that the original test set already included designed generalization test cases. For the same preference, we randomly sample scenes and objects when rendering these video demonstrations. To provide further insights, we conduct a set of additional experiments by intentionally generating cases where the demonstration and test videos are rendered in the same room with the same manipulated objects. This allows for a direct comparison of performance under consistent conditions. We test EILEV, LLaVA, and GPT-4 series models in this variant of PbP, as they have demonstrated relatively strong few-shot reasoning capability. See Table 4 for results.

A clear observation is that symbol-based reasoning (GPT-4) remains largely unaffected by differences in scenes or objects, while vision-based models are more susceptible to changes in the scene. This discrepancy can be attributed to the nature of our predefined preferences, which are high-level

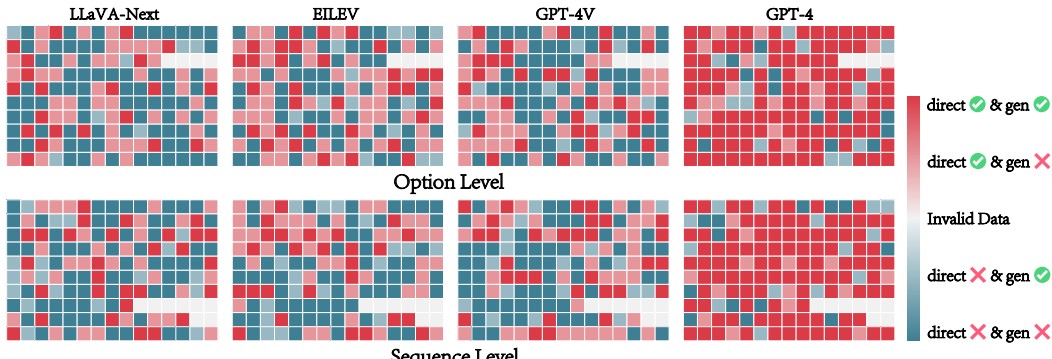

Figure 6: **Examination of the test points in both the *direct* and *gen* cases.** Each line denotes a separate scene. Different grid colors means different status of a datapoint.

and general enough to be applied across most scenes, regardless of new scenes or new objects. Conversely, the few-shot learned preferences in vision-based models are closely tied to specific visual cues associated with the scene or the objects. Consequently, when these visual elements change, the model's ability to recognize and apply the correct preferences could be compromised. This phenomenon, often referred to as contextual adaptability, has long been a challenge for vision-based models, which tend to overfit to specific scenes in the training videos.

We further plot test points and examine the prediction results in both the *direct* and *gen* cases, as shown in Figure 6. We made two key observations: 1) Preference learning is somehow related to the scenes; for certain scenes, models struggle to learn human preferences in both cases. 2) While models perform better in *direct* cases, the failure cases of the two settings are not completely repeated, especially in vision-based models. This indicates that models rely more on the contextual consistency of the visual environment to make accurate predictions, suggesting that they may not truly understand the preferences demonstrated in the video even when they predict the right preference. In general, symbol-based reasoning demonstrates robustness across diverse environments and objects, due to the high-level and general nature of predefined preferences. In contrast, vision-based models are more susceptible to changes in the visual context, as they rely heavily on specific visual cues associated with scenes or objects. This reliance on contextual consistency can hinder their ability to generalize effectively.

# 6 CONCLUSION

In this paper, we explore how embodied agents can learn and implement human preferences by observing human behaviors and interacting with human users. We introduce Preference-based Planning (PbP), a realistic embodied environment tailored to capture the diverse and complex aspects of human preferences in everyday life. Additionally, we establish an evaluation benchmark to assess the ability of various models to learn and utilize human preferences. Our experiments demonstrate that preference serves as a valuable abstraction of human behaviors and can guide subsequent planning efforts. Although inferring human preferences and planning actions that adapt to them from limited observations remains a considerable challenge for current models, incorporating preference into the reasoning and planning process enhances both effectiveness and generalizability. This is particularly true for symbol-based systems, which represent an idealized version of real-world settings. We hope our work will advance further research in this largely under-explored yet critical field of developing embodied agents capable of adapting to personalized needs and preferences.

**Limitations & Societal Impacts** The primary limitation of our work lies in the synthetic nature of the dataset. While the simulator we used, Omniverse, excels in rendering realistic scenes, there remains a gap in capturing the full complexity and variability of real-world settings. Additionally, the human-defined preference labels may not fully encapsulate the intricacies and diversity of human preferences. Moving forward, we are collecting preference demonstrations from real-world human daily life using head-worn devices, despite the considerable challenges involved. As preferences discussed in this paper are within a private scenario, we do not foresee any negative societal impacts stemming from our research.

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

## A  DATASET CARD

We follow the datasheet proposed in Gebru et al. (2021) for documenting our proposed PbP:

1. **Motivation**

   (a) **For what purpose was the dataset created?**
   The benchmark was created to evaluate existing learning agents on their ability to understand and adapt to various human preferences. Specifically, it aims to test the agents' proficiency in few-shot learning from demonstrations, where they must respond to ambiguous task instructions and formulate adaptive task plans based on limited examples of user preferences. The benchmark is designed to highlight the challenges and gaps in current AI systems' capabilities in planning activities and abstracting human preferences, ultimately driving advancements towards developing more intelligent and personalized embodied agents.

   (b) **Who created the dataset and on behalf of which entity?**
   N/A.

   (c) **Who funded the creation of the dataset?**
   N/A.

   (d) **Any other Comments?**
   None.

2. **Composition**

   (a) **What do the instances that comprise the dataset represent?**
   Each instance contains an egocentric video of an agent's activity, its bird's-eye-view map of the position of the agent, and a frame-level textual annotation of the current action, as shown in Figure 4. Additionally, we provide a rendered third-person view of the entire process.

   (b) **How many instances are there in total?**
   15000.

   (c) **Does the dataset contain all possible instances or is it a sample (not necessarily random) of instances from a larger set?**
   No. The dataset contains a set of demonstrations rendered within the simulator, The users can render more diverse instances if they want. We have provided the rendering instructions.

   (d) **What data does each instance consist of?**
   The instances that comprise the benchmark represent various types of human preferences applied to different tasks within a realistic embodied environment. Each instance is designed to challenge the learning agents to understand and adapt to these preferences based on a few demonstration examples, reflecting the diverse and hierarchical nature of user preferences in real-world scenarios. See above for data details.

   (e) **Is there a label or target associated with each instance?**
   Yes.

   (f) **Is any information missing from individual instances?**
   No.

   (g) **Are relationships between individual instances made explicit?**
   Yes.

   (h) **Are there recommended data splits?**
   No.

   (i) **Are there any errors, sources of noise, or redundancies in the dataset?**
   No.

   (j) **Is the dataset self-contained, or does it link to or otherwise rely on external resources (*e.g.*, websites, tweets, other datasets)?**
   Self-contained.

   (k) **Does the dataset contain data that might be considered confidential (*e.g.*, data that is protected by legal privilege or by doctor-patient confidentiality, data that includes the content of individuals' non-public communications)?**
   No.

(l) **Does the dataset contain data that, if viewed directly, might be offensive, insulting, threatening, or might otherwise cause anxiety?**
No.

(m) **Does the dataset relate to people?**
No.

(n) **Does the dataset identify any subpopulations (*e.g.*, by age, gender)?**
No.

(o) **Is it possible to identify individuals (*i.e.*, one or more natural persons), either directly or indirectly (*i.e.*, in combination with other data) from the dataset?**
No.

(p) **Does the dataset contain data that might be considered sensitive in any way (*e.g.*, data that reveals racial or ethnic origins, sexual orientations, religious beliefs, political opinions or union memberships, or locations; financial or health data; biometric or genetic data; forms of government identification, such as social security numbers; criminal history)?**
No.

(q) **Any other comments?**
None.

3. **Collection Process**

(a) **How was the data associated with each instance acquired?**
We render PbP using NVIDIA's Omniverse and OmniGibson simulation environment (Li et al., 2023).

(b) **What mechanisms or procedures were used to collect the data (*e.g.*, hardware apparatus or sensor, manual human curation, software program, software API)?**
The data for each instance in the benchmark was acquired by sampling preferences from a predefined set and constructing tasks paired with a few demonstrations that shared high-level preferences but differed in specific objects and scenes. Each sampled preference was randomly assigned to one of the 50 scenes provided by OmniGibson, with relevant objects sampled within the scene. Egocentric observation and action sequences of an embodied agent were generated as the agent performed tasks guided by a rule-based planner using planning primitives like inverse kinematics for grasping and the A* algorithm for movement.

(c) **If the dataset is a sample from a larger set, what was the sampling strategy (*e.g.*, deterministic, probabilistic with specific sampling probabilities)?**
N/A.

(d) **Who was involved in the data collection process (*e.g.*, students, crowdworkers, contractors) and how were they compensated (*e.g.*, how much were crowdworkers paid)?**
N/A.

(e) **Over what timeframe was the data collected?**
N/A.

(f) **Were any ethical review processes conducted (*e.g.*, by an institutional review board)?**
The dataset raises no ethical concerns.

(g) **Does the dataset relate to people?**
No.

(h) **Did you collect the data from the individuals in question directly, or obtain it via third parties or other sources (*e.g.*, websites)?**
N/A.

(i) **Were the individuals in question notified about the data collection?**
N/A.

(j) **Did the individuals in question consent to the collection and use of their data?**
N/A.

(k) **If consent was obtained, were the consenting individuals provided with a mechanism to revoke their consent in the future or for certain uses?**
N/A.

(l) **Has an analysis of the potential impact of the dataset and its use on data subjects (*e.g.*, a data protection impact analysis) been conducted?**
Yes.

(m) **Any other comments?**
None.

4. **Preprocessing, Cleaning and Labeling**

   (a) **Was any preprocessing/cleaning/labeling of the data done (*e.g.*, discretization or bucketing, tokenization, part-of-speech tagging, SIFT feature extraction, removal of instances, processing of missing values)?**
   N/A.

   (b) **Was the "raw" data saved in addition to the preprocessed/cleaned/labeled data (*e.g.*, to support unanticipated future uses)?**
   N/A.

   (c) **Is the software used to preprocess/clean/label the instances available?**
   N/A.

   (d) **Any other comments?**
   None.

5. **Uses**

   (a) **Has the dataset been used for any tasks already?**
   No, the dataset is newly proposed by us.

   (b) **Is there a repository that links to any or all papers or systems that use the dataset?**
   No, the dataset is new.

   (c) **What (other) tasks could the dataset be used for?**
   This dataset could be used for research topics like embodied AI and human-computer interaction.

   (d) **Is there anything about the composition of the dataset or the way it was collected and preprocessed/cleaned/labeled that might impact future uses?**
   N/A.

   (e) **Are there tasks for which the dataset should not be used?**
   N/A.

   (f) **Any other comments?**
   None.

6. **Distribution**

   (a) **Will the dataset be distributed to third parties outside of the entity (*e.g.*, company, institution, organization) on behalf of which the dataset was created?**
   No before it is made public.

   (b) **How will the dataset be distributed (*e.g.*, tarball on website, API, GitHub)?**
   On our project website upon acceptance.

   (c) **When will the dataset be distributed?**
   Upon acceptance.

   (d) **Will the dataset be distributed under a copyright or other intellectual property (IP) license, and/or under applicable terms of use (ToU)?**
   Under CC BY-NC [1] license.

   (e) **Have any third parties imposed IP-based or other restrictions on the data associated with the instances?**
   No.

   (f) **Do any export controls or other regulatory restrictions apply to the dataset or to individual instances?**
   No.

   (g) **Any other comments?**
   None.

7. **Maintenance**

---

[1] https://creativecommons.org/licenses/by-nc/4.0/

(a) **Who is supporting/hosting/maintaining the dataset?**
The authors.

(b) **How can the owner/curator/manager of the dataset be contacted (*e.g.*, email address)?**
N/A.

(c) **Is there an erratum?**
Future erratum will be released through the website.

(d) **Will the dataset be updated (*e.g.*, to correct labeling errors, add new instances, delete instances')?**
Yes.

(e) **If the dataset relates to people, are there applicable limits on the retention of the data associated with the instances (*e.g.*, were individuals in question told that their data would be retained for a fixed period of time and then deleted)?**
N/A. The dataset does not relate to people.

(f) **Will older versions of the dataset continue to be supported/hosted/maintained?**
Yes.

(g) **If others want to extend/augment/build on/contribute to the dataset, is there a mechanism for them to do so?**
Yes. We will release the source code as well as a licence on our project website after acceptance.

(h) **Any other comments?**
None.

# B  DATASET STATISTICS

The length of the simulations in dataset ranges from 1 to 5 minutes, depending on the tasks recorded. And the videos are recorded at 30 fps.

## B.1  PREFERENES

See Table A1 for the preference statistics in PbP.

Table A1: **Dataset Statistics in PbP.**

|  | Action_Level | Option_Level | Sequence_Level |
|---|---|---|---|
| **Preference Num** | 75 | 135 | 80 |
| **Video Num** | 5000 | 5000 | 5000 |
| **Sub-task Num** | 1 | 2-3 | 2-3 |

## B.2  ACTIONS

See Table A2 for the action statistics in PbP. We implement 17 action primitives in PbP to assist with model planning and dataset rendering. These action primitives have parameters that simplify tasks and are considered the lowest-level actions. Each sub-task contains 8 to 20 such lowest-level actions. Generally, most of these actions consist of two parts: the robot movement part and the arm (gripper) execution part. For robot movement, we use the A* algorithm to find paths and avoid collisions. We build a connection map during scene initialization for navigation, taking the robot's width into consideration. For the arm (gripper) execution, we primarily use the IK algorithm to compute arm movements. However, since IK cannot handle complex tasks, such as picking objects from the fridge, we also leverage the Open Motion Planning Library (OMPL) planner (Sucan et al., 2012) with forward planning to assist in planning the arm positions.

## B.3  MORE DATASET DETAILS AND DISCUSSION

**Dataset production**  The process of producing data is mainly explained in Section 3.2. In summary, we follow the order of "sample preference - sample scene - sample objects to be manipulated - generate actions guided by a rule-based planner."

**Length and FPS of the simulations** The length of the simulations ranges from 1 to 5 minutes, depending on the tasks recorded. The videos are recorded at 30 fps.

**Actions contained in each simulation** The number of actions in simulations varies among different preference levels. There is 1 subtask for action-level, 2-3 subtasks for option-level, and 2-3 subtasks for sequence-level preferences. Each subtask contains 8-20 actions.

**Scenes and rooms** Each scene contains various types of rooms. The main differences between scenes are the type, number, and layout of both rooms and furniture. Additionally, each room may contain different objects and have unique layouts. Details of the scenes and rooms can be found in Omnigibson's official documentation (https://behavior.stanford.edu/omnigibson/), as we directly adopt these scenes from the open-sourced project.

**290 preference types** Considering that preferences in household activities are not only multi-dimensional but also hierarchical, we first define a hierarchy of preferences from the perspective of how things happen in a life scenario, that is, from each specific action to a sub-task consisting of several actions, and then to the sequence combining these sub-tasks. The next step is to expand each level with typical tasks and actions. The detailed definition of the 290 preferences can be found in Section 3.1.

**The egocentri view** Collecting both egocentric observations and third-person views is feasible in PbP or similar environments built on simulators like iGibson. However, in real-world scenarios, it is generally easier to gather egocentric observations of human daily activities, as these can be efficiently captured through wearable devices. Additionally, there are numerous egocentric-view datasets available, such as Ego4D(Grauman et al., 2022), which further facilitate this approach. While third-person views can provide a different perspective, they often encounter issues such as occlusion. Although research based on third-person views is essential for applications involving real robots, focusing on egocentric views in the current work allows for a more straightforward exploration of preference learning and planning. Nevertheless, third-person view data can be obtained by integrating additional cameras, as outlined in our provided code.

**Action ground truth** In experiments involving vision input, we do not explicitly provide the action sequence of the user. In symbolic-based experiment, we provide the action sequence to reduce the perception cost to concentrate more effectively on the inference and planning aspects of the study.

Table A2: Action Primitives in PbP.

| Action List | Explanation |
|---|---|
| **Move_to_[]** | Move to a specified location, or a specified room, or a specified object |
| **Rotate_to_[]** | Rotate to a specified orientation or a specified object |
| **Pick_[]** | Pick up an object using the gripper, *e.g.*, "Pick_apple" |
| **Place_[]** | Place an object at a location, *e.g.*, "Place_apple_on_table" |
| **Fill_[]_with_[]** | Fill a container with a substance, *e.g.*, "Fill_glass_with_water" |
| **Pour_[]** | Pour a substance from a container, *e.g.*, "Pour_milk" |
| **Open_[]** | Open an object, *e.g.*, "Open_door" |
| **Close_[]** | Close an object, *e.g.*, "Close_fridge" |
| **Cut_[]** | Cut an object, *e.g.*, "Cut_carrot" |
| **Cook_[]** | Cook an item, *e.g.*, "Cook_pasta" |
| **Wash_[]** | Wash an object, *e.g.*, "Wash_dishes" |
| **Clean_[]** | Clean a surface or object, *e.g.*, "Clean_counter" |
| **Cover_[]** | Cover an object, *e.g.*, "Cover_bowl" |
| **Uncover_[]** | Uncover an object, *e.g.*, "Uncover_bowl" |
| **Toggle_on_[]** | Turn on a device, *e.g.*, "Toggle_on_light" |
| **Toggle_off_[]** | Turn off a device, *e.g.*, "Toggle_off_stove" |
| **Wait_[]** | Wait some time |

## C  Experiment Details

### C.1  Case Study

We also provide a case with preference *Put fruit on the bed* in the following table. We present a simplified version of the demonstrations, where all video outputs have been translated into symbol-based action sequences for ease of understanding. Video-based models such as LLaVA-Next and GPT-4V struggle with comprehending preferences and tend to replicate certain action patterns from the video demonstration, such as "move to" and "pick up." Llama3 demonstrates a partial understanding and execution of the preference. It correctly moves to each fruit (grape, banana), picks them up, and places them on the bed. However, it also interacts with the pencil and places it on the bed, which is not required by the preference. Ideally, the pencil should be placed on the table, similar to the pen. On the other hand, GPT-4(Symbol) accurately interacts with the grape and banana by moving to each fruit, picking it up, and placing it on the bed. This demonstrates a better understanding and execution of the preference compared to the other models.

| Case | ID | Scene | Actions |
|---|---|---|---|
| **Context** | Demo 1 | Beechwood | move to orange, pick up orange, move to bed with orange, place orange on bed, move to apple, pick up apple, move to bed with apple, place apple on bed, move to dvd, pick up dvd, move to table with dvd, place dvd on table. |
| | Demo 2 | Rs | move to banana, pick up banana, move to bed with banana, place banana on bed, move to club sandwich, pick up club sandwich, move to countertop with club sandwich, place club sandwich on countertop, move to orange, pick up orange, move to countertop with orange, move to bed with orange, place orange on bed. |
| | Demo 3 | Merom | move to pen, pick up pen, move to table with pen, place pen, move to strawberry, pick up strawberry, move with to bed with strawberry, place strawberry on bed, move to textbook, pick up textbook, move with textbook, move to bookshelf with textbook, place textbook in bookshelf. |
| **Test Scene** | | Beechwood | grape, pencil, banana |
| **Answer** | LLaVA-Next | Beechwood | move to grape, move to bed with grape, move to pencil, pick up pencil, move to banana, pick up banana |
| | GPT-4V | Beechwood | move to grape, pick up grape, move to bed with grape, place grape on bed |
| | Llama3 | Beechwood | move to grape, pick up grape, move to bed with grape, place grape on bed, move to pencil, pick up pencil, move to bed with pencil, place pencil on bed, move to banana, pick up banana, move to bed with banana, place banana on bed |
| | GPT-4(Symbol) | Beechwood | move to grape, pick up grape, move to bed with grape, place grape on bed, move to banana, pick up banana, move to bed with banana, place banana on bed |

Table A3: Case Study with preference *Put fruit on the bed*.

## D  Baseline Details

### D.1  ViViT

Inspired by Vision Transformer, ViViT extracts spatio-temporal tokens from the input video and outputs video classification labels for classification. We adopt the ViViT implementation from the official github repo https://github.com/google-research/scenic.

Specifically, we utilize a ViViT with an image size of 224 and a patch size of 16. We extract 2 frames per second from the input video and pad them with the last frame. The Transformer architecture with 3 attention heads operates on features of hidden size of 192 and depth of 4. Each attention head

operates on a dimension of 64. We train our model for 30 epochs with a learning rate 3e-5. For the few-shot setting, we concatenate the demo videos temporally.

## D.2    LLAVA

Following the official implementation of LLaVA from https://github.com/LLaVA-VL/LLaVA-NeXT, we test the LLaVA-NeXT-Video-7B-DPO model which is designed for video understanding. Specifically, we run the model following the default inference settings, with vicuna_v1 as the prompt mode, a sample frame number of 32, and a spatial pooling stride of 2. The textual prompts are as follows[2]:

```
"Stage One / Preference Prediction"
You are a robot assistant that can help summarize the host's preference.
All possible preferences are: {ALL POSSIBLE PREFERENCES}
Now there are some prevous video demos:
[VIDEO_DEMO_1] The preference is [PREFERENCE_1]
[VIDEO_DEMO_2] The preference is [PREFERENCE_2]
[VIDEO_DEMO_3] The preference is [PREFERENCE_3]
Now, please summarize the preference from the last video: [TEST_CASE]
Quesiton: What's the user's preference? Choose from the preference listed before:

"Stage Two / Planning"
You are a robot assistant. Please view the demos and help generate action sequence.
All possible preferences are: {ALL POSSIBLE ACTIONS}
Now there are some prevous video demos:
[VIDEO_DEMO_1]
[VIDEO_DEMO_2]
[VIDEO_DEMO_3]
Now you are in the scene with [SCENE DESCRIPTIONS]. Your action sequence is:
```

## D.3    EILEV

Following the official implementation from https://github.com/yukw777/EILEV.git, we test the EILEV model in PbP. There are two reasons we chose EILEV among other VLMs as one of our baselines: 1) EILEV elicits in-context learning through a series of architectural modifications and a unique training process, 2) EILEV is trained using ego-centric data, which is compatible with PbP's input. The textual prompts are as follows. Since EILEV requires the input of the videos and texts to follow a certain pattern for better in-context learning, there are some small modifications to the prompt:

```
"Stage One / Preference Prediction"
You are a robot assistant that can help summarize the host's preference.
All possible preferences are: {ALL POSSIBLE PREFERENCES}
Quesiton: What's the user's preference? Choose from the preference listed before:
Now there are some prevous video demos:
[VIDEO_DEMO_1] The preference is [PREFERENCE_1]
[VIDEO_DEMO_2] The preference is [PREFERENCE_2]
[VIDEO_DEMO_3] The preference is [PREFERENCE_3]
[TEST_CASE]

"Stage Two / Planning"
You are a robot assistant. Please view the demos and help generate action sequence.
```

---

[2]For the textual prompts, we aim to maintain consistency across all LLM models, although some baselines may have additional requirements for the input format. The prompt design is mainly motivated by OpenAI Cookbook git@github.com:openai/openai-cookbook.git. We omitted the prompt tuning process, as we found that minor changes in the prompt were unlikely to significantly impact the results. Conversely, selecting the proper demonstrations in the few-shot examples has a much greater influence on the results.

```
All possible preferences are: {ALL POSSIBLE ACTIONS}
Now there are some prevous video demos:
[VIDEO_DEMO_1]
[VIDEO_DEMO_2]
[VIDEO_DEMO_3]
Now you are in the scene with [SCENE DESCRIPTIONS]. Your action sequence is:
```

### D.4 GPT-4V

We run our GPT-4 model through the AzureOpenAI API using the GPT version "gpt-4-turbo-2024-04-09". The API has a limit of 10 images per request. Consequently, for the zero-shot setting, we resample each input video to 8 frames of size 224. For the few-shot setting, where we need to input 3 extra video demonstrations, we concatenate 4 images into a frame, thereby obtaining 4 videos in 8 frames, maintaining the same frame number as the previous setting. We test the model with a temperature of 0.05. The textual prompts are as follows:

```
"Stage One / Preference Prediction"
You are a robot assistant that can help summarize the host's preference.
All possible preferences are: {ALL POSSIBLE PREFERENCES}
Now there are some prevous video demos:
[VIDEO_DEMO_1] The preference is [PREFERENCE_1]
[VIDEO_DEMO_2] The preference is [PREFERENCE_2]
[VIDEO_DEMO_3] The preference is [PREFERENCE_3]
Now, please summarize the preference from the last video: [TEST_CASE]
Quesiton: What's the user's preference? Choose from the preference listed before:

"Stage Two / Planning"
You are a robot assistant. Please view the demos and help generate action sequence.
All possible preferences are: {ALL POSSIBLE ACTIONS}
Now there are some prevous video demos:
[VIDEO_DEMO_1]
[VIDEO_DEMO_2]
[VIDEO_DEMO_3]
Now you are in the scene with [SCENE DESCRIPTIONS]. Your action sequence is:
```

### D.5 DAG-OPT

We implement the DAG-Opt baseline following https://github.com/xunzheng/notears.git. Specifically, we implement a nonlinear NOTEARS using MLP in evaluation.

### D.6 LLAMA3-8B

We test the Llama3 series model with the official scripts from https://github.com/meta-llama/llama3. Specially, we test the 8B instruction-tuned variant "Meta-Llama-3-8B-Instruct" on PbP. We test the model with a temperature of 0.05. The textual prompts are as follows:

```
"Stage One / Preference Prediction"
You are a robot assistant that can help summarize the host's preference.
Please read the following text file and summarize the user's preference.
All possible preferences are: {ALL POSSIBLE PREFERENCES}
[TEXT_ANNOTATION_1] The preference is [PREFERENCE_1]
[TEXT_ANNOTATION_2] The preference is [PREFERENCE_2]
[TEXT_ANNOTATION_3] The preference is [PREFERENCE_3]
Now, please summarize the preference from the last tet file: [TEST_CASE]
Quesiton: What's the user's preference? Choose from the preference listed before:

"Stage Two / Planning"
You are a robot assistant. Please read the following text files and help generate action sequence.
```

```
All possible preferences are: {ALL POSSIBLE ACTIONS}
Now there are some prevous video demos:
[TEXT_ANNOTATION_1] (action sequence)
[TEXT_ANNOTATION_2] (action sequence)
[TEXT_ANNOTATION_3] (action sequence)
Now you are in the scene with [SCENE DESCRIPTIONS]. Your action sequence is:
```

## D.7  GPT-4

We use "gpt-4-turbo-2024-04-09" with a temperature of 0.05. The textual prompts are as follows:

```
"Stage One / Preference Prediction"
You are a robot assistant that can help summarize the host's preference.
Please read the following text file and summarize the user's preference.
All possible preferences are: {ALL POSSIBLE PREFERENCES}
[TEXT_ANNOTATION_1] The preference is [PREFERENCE_1]
[TEXT_ANNOTATION_2] The preference is [PREFERENCE_2]
[TEXT_ANNOTATION_3] The preference is [PREFERENCE_3]
Now, please summarize the preference from the last tet file: [TEST_CASE]
Quesiton: What's the user's preference? Choose from the preference listed before:
```

```
"Stage Two / Planning"
You are a robot assistant. Please read the following text files and help generate action sequence.
All possible preferences are: {ALL POSSIBLE ACTIONS}
Now there are some prevous video demos:
[TEXT_ANNOTATION_1] (action sequence)
[TEXT_ANNOTATION_2] (action sequence)
[TEXT_ANNOTATION_3] (action sequence)
Now you are in the scene with [SCENE DESCRIPTIONS]. Your action sequence is:
```

