# OpenReview forum: "Learning to Plan with Personalized Preferences"
_ICLR.cc/2025/Conference — Submitted to ICLR 2025_

### Official Review · Reviewer_nfLL · 2024-11-04

**Soundness:** 3
**Presentation:** 3
**Contribution:** 3
**Rating:** 8
**Confidence:** 3

**Summary:**

The paper attempts towards building a benchmark/environment for building more personalized agents. In this paper, preferences ext beyond the task of rearrangement (usual in prior works) and the paper contains stronger definitions of what preferences mean.

They introduce a new embodied environment PbP which supports several preferences at various levels (action - how to handle specific items, option - which items to prefer, sequence - preference on order of tasks), and various range of tasks (not just limited to  preferences in rearrangement like previous works).

The authors use Omniverse/OmniGibson environments/scenes with 50 scenes, with the Fetch robot as their agent in example. The reference dataset is the the Behavior-1K dataset and they build a preference vocabulary with 290 different preferences.

They generate some demonstrations once a preference and scene is sampled using oracle algorithms. The input is then as a top-down map, egocentric view, textual annotation of the current action, and third-person view of the agent. The problem is modeled as few-shot LfD.

The experiments are performed on vision-models (ViViT) VLMs (LLaVA, EILEV, GPT-4V), LLMs (GPT-4), and Symbol-Based models (GPT-4, LLaMa3).  The evaluation is done using Levenshtein distance.

They perform end-to-end and two-stage learning-planning experiments. They find that training the model to predict the preference first is better than end-to-end action prediction.

They show that VLMs struggle with learning preferences compared to LLMs, and also contain ablation study where they test on the generalization ability of their models to different scenes and objects, and same scenes and objects as the demonstration. GPT-4 text model models seem most robust, followed by GPT-4V towards generalization. They also provide an insight that the vision-based models rely on visual context to model preferences and are less robust in comparison.

**Strengths:**

- The paper introduces a structured approach and definitions towards modeling preferences which is a highly novel contribution.
- In order to support building agents to learn preferences, they provide a new environment and tasks which could serve as a useful benchmark for evaluating personalization.
- They perform useful ablations and add a generalization vs direct experiment which provide an understanding of how robust the baselines are towards novel scenes/objects.
- The figures are very helpful in understanding the main ideas and the approach of the paper.

**Weaknesses:**

- The preference vocabulary adds limitations to the paper, as it definitely will not cover all possible preferences possible in the real-world. I have doubts about the scalability of the approach to an "open-preference-vocab" scenario. Would love to hear the authors thoughts on this.
- They pass in various kinds of image-based observations - are all these observations necessary? For example, would an agent still work as well if the third-person view is not provided? This ablation seems to be missing.
- Their text-only baselines perform significantly better.  While the argument that perception models are just not good enough might be valid, this could also means that the visual cues are not necessary for modeling their  preferences and add unnecessary observations. This probably needs some investigation or refuting argument in the paper.
- One ablation that is missing is how well do they two-stage models perform when GT preference is provided.
- Minor fixes:
	- Line 183 : "whether them" -> "whether they can"
	- Figure 4: This is a nit but, the top-down map seems to be flipped in comparison to the 3D view. Shouldn't affect the agent, but might help for things to be consistent.
	- Line 345: "socre-based" -> "score-based"

**Questions:**

How do the authors propose to handle cases when preference for one specific item (say apples) will vary from that of another specific item (say pears) and that is not provided in the demonstrations? Do the evaluations also have this component?
- Table 1/2:  Why are there no results on the action-level data?
- Second stage results are always better, and GPT-4 seems to be the best, does that mean actions are key to understanding preferences?

---

> ### Author Response · Authors · 2024-11-21
>
> Dear Reviewer nfLL,
>
> We sincerely appreciate your insightful and constructive feedback on our paper.
>
> > The preference vocabulary adds limitations to the paper, as it  definitely will not cover all possible preferences possible in the  real-world. I have doubts about the scalability of the approach to an  "open-preference-vocab" scenario. Would love to hear the authors  thoughts on this.
>
> Thank you for your valuable advice. Yes, the preference definitions in our work are pre-defined by humans, based on previous works in the community, especially Behavior-1k [1]. Actually, these preferences can be seen as high-level abstractions of action sequences. The generalization of our preference modeling doesn't necessarily depend on the scope of these pre-defined preference definitions. The models also learn from action sequences and finally output actions as well, as we shown in Sec 5.2.
> The pre-defined preferences mainly serve as a guide to help us sample demonstrations and help model planning. While they may not cover all corner cases, they provide a substantial enough range to serve as a benchmark to evaluate the baselines. Our proposed three-tiered hierarchical structure of preferences is designed from the perspective of how things happen in a life scenario. This structure progresses from each specific action to a sub-task consisting of several actions, and then to the sequence combining these sub-tasks. This design helps the model to plan in a hierarchical way for generalization.
> Besides, so long as the task can be formulated as action sequences and sub-structures exist in the action distributions, the methodology can be applied, no matter how domains vary.
> [1] Li, C., Zhang, R., Wong, J., Gokmen, C., Srivastava, S., Martín-Martín, R., Wang, C., Levine, G., Lingelbach, M., Sun, J., et al. (2023). Behavior-1k: A benchmark for embodied ai with 1,000 everyday activities and realistic simulation. In Conference on Robot Learning (CoRL).
>
> > - They pass in various kinds of image-based observations - are all  these observations necessary? For example, would an agent still work as  well if the third-person view is not provided? This ablation seems to be  missing.
>
> We have included third-person vision observations in the dataset for better illustration, but not in experiments. We only choose the egocentric view of the agent in the simulator as the main input in the dataset for two primary reasons: 1) the egocentric perspective provides a clear view with minimal occlusions, and 2) it mimics a human's view, making the dataset and models easily transferable to real-world data collected by head-worn devices. Please refer to the second paragraph of sec 3.2.
>
> > - Their text-only baselines perform significantly better.  While the  argument that perception models are just not good enough might be valid,  this could also means that the visual cues are not necessary for modeling their  preferences and add unnecessary observations. This  probably needs some investigation or refuting argument in the paper.
>
> Visual observations are utilized to align with real-world scenarios, where textual descriptions of action sequences are rarely available, and vision is the most common method for information acquisition. Using text to feed models is indeed a simplification of the task proposed in our paper, allowing us to minimize difficulties at the perception level and concentrate more on the reasoning and planning aspects of the task. However, we argue that comparing the performance of vision-based and symbol-based baselines is essential, as visual observations represent the settings models typically encounter in real-world scenarios. Furthermore, although visual cues may introduce some noise into the process of understanding and planning, as you suggested, they also provide richness and depth of context not typically available through textual data alone. Visual cues can offer nuanced insights into user preferences that are not explicitly stated, making them invaluable for applications where understanding subtle, non-verbal user behaviors is critical. Therefore, while text-only baselines may perform better in our current settings, we believe that experiments with visual inputs should be maintained.
>
> > One ablation that is missing is how well do they two-stage models perform when GT preference is provided.
>
> Thank you for your valuable suggestion. The ablation you mentioned pertains more to the aspect of task planning based on GT preferences. While this is not the central focus of our current paper and it takes a long time to test all models, we are running the experiments as soon as possible and will address it in future revisions.
>
> > Minor fixes
>
> We really appreciate your thoughtful reading and practical suggestions. We have checked and modified them one by one in revision.

---

> > ### Comment · Reviewer_nfLL · 2024-11-27
> > **Response to rebuttal**
> >
> > > The generalization of our preference modeling doesn't necessarily depend on the scope of these pre-defined preference definitions. The models also learn from action sequences and finally output actions as well, as we shown in Sec 5.2.
> >
> > > The pre-defined preferences mainly serve as a guide to help us sample demonstrations and help model planning. While they may not cover all corner cases, they provide a substantial enough range to serve as a benchmark to evaluate the baselines... This structure progresses from each specific action to a sub-task consisting of several actions, and then to the sequence combining these sub-tasks.
> >
> > Thank you so much for the clarification. I agree that since this is based on a prior work, and it has diverse enough preferences, it serves as a valid benchmark to evaluate the approach.
> >
> > > We only choose the egocentric view of the agent in the simulator as the main input in the dataset for two primary reasons: 1) the egocentric perspective provides a clear view with minimal occlusions, and 2) it mimics a human's view, making the dataset and models easily transferable to real-world data collected by head-worn devices. Please refer to the second paragraph of sec 3.2.
> >
> > Thanks for clarifying this, sorry for missing this in the reading of the paper.
> >
> > > Furthermore, although visual cues may introduce some noise into the process of understanding and planning, as you suggested, they also provide richness and depth of context not typically available through textual data alone.
> >
> > I agree with this point.
> >
> > > Therefore, while text-only baselines may perform better in our current settings, we believe that experiments with visual inputs should be maintained.
> >
> > I never suggested that the experiments with visual inputs should be removed, just that if it is possible to evaluate how much of the drop is due to additional noise/extra information from the visual cues. Infact, I like the experiments with visual input and they provide significant value.
> >
> > > Thank you for your valuable suggestion. The ablation you mentioned pertains more to the aspect of task planning based on GT preferences. While this is not the central focus of our current paper and it takes a long time to test all models, we are running the experiments as soon as possible and will address it in future revisions.
> >
> > Thank you so much for addressing this, looking forward to the results in the paper.
> >
> > Overall, I like the paper, and the weaknesses I shared were minor or non-existent based on the authors' rebuttal. I shall retain my rating as it is a valuable contribution.

---

> ### Author Response · Authors · 2024-11-21
>
> > How do the authors  propose to handle cases when preference for one specific item (say  apples) will vary from that of another specific item (say pears) and  that is not provided in the demonstrations? Do the evaluations also have  this component?
>
> Yes, our dataset does cover these cases. For scenarios that are specific to a certain action or object, if they are not represented in the demonstrations, the model will not be aware of these preferences. Consequently, they will not be included in the evaluation sampling.
>
> > Table 1/2:  Why are there no results on the action-level data?
>
> In the given context, action preferences are related to specific actions, such as placing an apple in a certain part of a shelf. These preferences are simple to perceive and copy, which means that a basic imitation policy can effectively solve the problem. We would like to focus on more complex preferences that cannot be easily addressed through simple copy/imitation, and to test models in scenarios where preferences involve more nuanced or context-dependent decisions.
>
> > Second stage results are always better, and GPT-4 seems to be the  best, does that mean actions are key to understanding preferences?
>
> Yes, from many perspective, actions are the most crucial element in understanding preferences, as they are the fundamental components reflecting human preferences, which are essentially high-level abstractions of action sequences. While in our paper, improved results in the second stage further demonstrate that, compared to end-to-end learning, explicitly modeling human preferences based on few-shot observations and then conducting further planning can enhance the models' understanding of human behaviors and yield more appropriate actions. This underscores the significance of preferences in the entire process.

---

> ### Author Response · Authors · 2024-11-30
>
> We greatly appreciate your valuable feedback on the rebuttal.
>
> Thank you again for your time and your thoughtful advice. Your kindness and insights help us improve our work and encourage us to continuously contribute to the field.

---

### Official Review · Reviewer_SCvY · 2024-11-04

**Soundness:** 3
**Presentation:** 3
**Contribution:** 2
**Rating:** 5
**Confidence:** 3

**Summary:**

This paper introduces the Preference-based Planning: PbP environment, used for training embodied AI agents to learn and adapt to individual human preferences in household settings, in hope to generalize to more diverse scenarios. PbP constructed a hierarchical preference structure across three levels: action, option, and sequence, within a realistic simulation (based on Omniverse and OmniGibson) and the Behavior-1k activities in 50 scenes. The authors evaluate various models in few-shot preference learning and preference-based planning, highlighting the performance gaps and challenges in modeling human preferences for adaptive action planning. The paper asserts that the inclusion of preference learning in planning could enhance embodied agents’ ability to handle personalized, context-sensitive tasks.

**Strengths:**

- Originality: The paper introduces a unique PbP environment to challenge current models when integrating personalized preferences into embodied AI planning.

- Quality: The structured design of PbP, including a comprehensive dataset of diverse (thousands of activities and sample objects items diversly), parameterized preferences, is well-executed and supports rigorous evaluation. The study includes experiments comparing the performance of both video-based and symbol-based models, in both end-to-end and two stage settings, in learning and applying preferences.

- Clarity: The task formulation for PbP is well written and easy to follow.

- Significance: The insights into the current limitations of multimodal models (e.g., GPT-4V, LLaVA) in inferring human preferences is valuable for future model development in embodied AI planning with personalization.

**Weaknesses:**

Although the paper has a great catch point of personalization in embodied AI planning, created a realistic environment, the PbP environment generation and the paper has following concerns:

- Implicit preference ambiguity: while the hierarchical structure (action, option, and sequence) aims to maintain consistency, the authors didn't adequately ensure that preferences are truly implicit and understood across different scenes. Variability in scene context and object interactions could lead to unintended changes in how a preference is perceived for same person. Clarifying how consistency is validated across scenes would strengthen the methodology.

- Biases in few-shot demonstrations? The reliance on rule-based generation of few-shot examples might introduce potential biases. The scenarios may be constructed in a way that fits the developers' understanding of user preferences, which might not capture the diversity and unpredictability of real-world user behaviors.

- The focus on few-shot learning alone limits the benchmark. Adding fine-tuned policy models for learnt personalized embodied planning such as fine-tuning Video-based models / symbolic based models with Monte Carlo Tree Search for trajectory generation, will largely strengthen the comprehensiveness of the PbP benchmark.

- The ablation study could have explored how each hierarchical preference level (action, option, sequence) specifically impacts the models’ learning and planning performance. This would help in understanding which levels are most challenging or influential for preference learning.

**Questions:**

Could the authors elaborate on the measures taken to ensure consistency and generalizability of user preferences across different scenes and contexts in the few-shot experiments?

Would additional ablation studies on the impact of varying the number of demonstrations on model performance enhance understanding of personalized planning capabilities? Would incorporating experiments involving fine-tuned models with additional user preference data improve the models' inference capabilities?

---

> ### Author Response · Authors · 2024-11-21
>
> Dear Reviewer SCvY,
>
> We sincerely appreciate your insightful and constructive feedback on our paper.
>
> > - Implicit preference ambiguity: ... Variability in scene context and  object interactions could lead to unintended changes in how a preference is perceived for same person. Clarifying how consistency is validated across scenes would strengthen the methodology.
> > - Biases in few-shot demonstrations? The  scenarios may be constructed in a way that fits the developers'  understanding of user preferences, which might not capture the diversity  and unpredictability of real-world user behaviors.
>
> We do agree with the two concerns raised, and acknowledge their importance for further research. However, our study has a different focus. The preference definitions in our work are pre-defined by humans, based on prior research in the community, particularly Behavior-1k [1]. These preferences can be viewed as high-level abstractions of action sequences. The generalization of our preference modeling in real-world scenarios does not solely depend on the scope of these pre-defined preferences. Our models also learn from action sequences and ultimately output actions, as demonstrated in Section 5.2. The pre-defined preferences primarily serve as a guide to facilitate the sampling of demonstrations and assist in model planning. Therefore, the bias from pre-defined preferences does not impact the task itself. While these preferences may not cover all corner cases, they provide a sufficiently broad range to serve as a benchmark for evaluating baselines in reasoning or planning. Our proposed three-tier hierarchical structure of preferences is designed to somehow reflect real-life preferences. This structure progresses from specific actions to a sub-task comprising several actions, and then to a sequence that combines these sub-tasks. This design enables to plan hierarchically, enhancing generalization.
> We did not ground the task in more challenging settings involving real humans, whose preferences might change unpredictably and involve complex real-world scenarios. Instead, we focused on fundamental reasoning and planning tasks where a stable preference can guide the entire process. However, moving forward, it is indeed possible that human-defined preference labels may not fully capture the diversity and unpredictability of real-world user behaviors. Adapting the pipeline to real-world scenarios could be a valuable direction for future work.
> [1] Li, C., Zhang, R., Wong, J., Gokmen, C., Srivastava, S., Martín-Martín, R., Wang, C., Levine, G., Lingelbach, M., Sun, J., et al. (2023). Behavior-1k: A benchmark for embodied ai with 1,000 everyday activities and realistic simulation. In CoRL.
>
> > - The focus on few-shot learning alone limits the benchmark. Adding fine-tuned policy models for learnt personalized embodied planning such  as fine-tuning Video-based models / symbolic based models with Monte  Carlo Tree Search for trajectory generation, will largely strengthen the  comprehensiveness of the PbP benchmark.
>
> We respectfully disagree. The formulation of preference learning as a few-shot problem is based on two considerations: i) Humans, even infants are found to have the ability to detect others' preference with only a few demonstrations. So to facilitate embodied AI, it is necessary to test such few-shot ability. ii) In a home assistant setting, it is infeasible to collect a large labeled set for a specific person and task. So in a realistic setting, we believe it is necessary to pay more attention to the few-shot setting.

---

> ### Author Response · Authors · 2024-11-21
>
> > The ablation study could have explored how each hierarchical  preference level (action, option, sequence) specifically impacts the  models’ learning and planning performance. This would help in  understanding which levels are most challenging or influential for  preference learning.
>
> We have conducted experiments about models' ability on both option and sequence level of preferences, and compared to analyze possible reasons behind the performance difference. Please refer to Sec.5 for details.
>
> > Could the authors elaborate on the measures taken to ensure consistency and generalizability of user preferences across different scenes and contexts in the few-shot experiments?
>
> We kindly refer you to the rebuttals provided for the first question. We are not addressing this issue in the current work.
>
> > Would additional ablation studies on the impact of varying the number  of demonstrations on model performance enhance understanding of  personalized planning capabilities? Would incorporating experiments  involving fine-tuned models with additional user preference data improve  the models' inference capabilities?
>
> Thank you for your valuable advice. We have conducted a small-scale ablation study concerning the number of demonstrations provided to the VLM models, as illustrated in the table below. Generally, our findings indicate that an increased number of demonstrations enhances the model's ability to learn human preferences and aids in further planning. This conclusion is particularly apparent in the Second-stage planning phase, where the model more accurately imitates the human's action sequence and achieves a lower sequence distance. Regarding the first-stage results, we observe that, in some cases, such as with the 5-demo-case of GPT-4 and EILVE, an excess of information can lead to confusion in the initial prediction stage. However, overall, the results align with our intuition.
> Regarding the inclusion of fine-tuned models with additional user preference data, we believe this will enhance the models' in-domain performance. However, there is a potential risk of overfitting, which could affect their ability to generalize. Additionally, please refer to the previous responses for an explanation of why we did not focus on fine-tuning baselines.
>
> ### First-Stage Results (accuracy, larger is better)
>
> | Model  | Metric        | 1     | 2     | 3 (in the paper) | 5     |
> |--------|---------------|-------|-------|------------------|-------|
> | **GPT4**    | Option Level  | 56%   | 74%   | 86.27%           | 69%   |
> |        | Sequence Level| 4%    | 80%   | 68.42%           | 83%   |
> | **Llama3** | Option Level  | 28%   | 37%   | 72.98%           | 82%   |
> |        | Sequence Level| 18%   | 32%   | 67.18%           | 67%   |
> | **EILEV**  | Option Level  | 12%   | 25%   | 36.87%           | 39%   |
> |        | Sequence Level| 12%   | 13%   | 24.85%           | 22%   |
>
> ### Second-Stage Results (distance, smaller is better)
>
> | Model  | Metric        | 1          | 2          | 3 (in the paper) | 5          |
> |--------|---------------|------------|------------|------------------|------------|
> | **GPT4**    | Option Level  | 1.56±2.25  | 0.25±0.85  | 0.12±3.12        | 0.12±0.59  |
> |        | Sequence Level| 17.74±7.50 | 12.52±8.71 | 12.29±3.12       | 10.6±7.63  |
> | **Llama3** | Option Level  | 12.57±3.43 | 8.78±5.31  | 8.22±5.58        | 4.12±2.02  |
> |        | Sequence Level| 27.57±8.32 | 24.23±5.35 | 19.02±7.10       | 12.12±6.78 |
> | **EILEV**  | Option Level  | 24.12±6.37 | 22.95±8.12 | 11.18±4.20       | 8.76±3.02  |
> |        | Sequence Level| 29.87±8.19 | 26.78±9.42 | 26.57±12.21      | 22.12±9.85 |

---

> ### Author Response · Authors · 2024-11-30
>
> Dear Reviewer, thank you again for your time and thoughtful review. Your valuable feedback has been instrumental in helping us refine and enhance our paper.
>
> As the discussion period comes to a close, we would be more than happy to provide additional explanations if you have any concerns or require further clarification.

---

### Official Review · Reviewer_rU8v · 2024-11-04

**Soundness:** 2
**Presentation:** 2
**Contribution:** 2
**Rating:** 3
**Confidence:** 3

**Summary:**

The paper introduces a framework for enabling embodied AI agents to adapt their behavior to individual human preferences. The authors propose the "Preference-based Planning" (PbP) environment, built upon OmniGibson, which simulates a diverse set of user preferences in embodied setting. The paper presents a comprehensive evaluation using state-of-the-art models, analyzing their ability to infer preferences through few-shot learning and apply these preferences in dynamic planning tasks.

**Strengths:**

1) Given the increasing interest in LLM based agents, personalization preference learning is becoming an important problem.
2) The baselines used in the paper are extensive and provide a good understanding of the current state of the area.

**Weaknesses:**

1) Given that the symbol based LLMs work so well compared to the vision based methods, it undermines the utility of the benchmark for preference learning from visual inputs. The example task in Table A.3 gives you an idea about this. I was able to understand the preference of the user for that task without having access to the video.
2) Some of assumptions made at the task level do not make sense to me:
  a) How would the agent get access to egocentric observations of the user? Ideally the robot should learn to pick up preferences by watching a human perform the task from its own camera.
  b) What embodiment is used during the preference demonstrations?
  c) Similarly having access to the action sequence of the user.
  It would be good to see a discussion around these questions in the paper.
3) The decision of using Levenshtein distance as the main metric for the evaluation is arbitrary and penalises the agent for not exactly matching the training data format. Given that the authors have access to a simulator, I would have liked seeing the evaluation of the actions in the simulator.
4) Figure 2 can be improved:
  a) The text formatting is poor.
  b) The example tasks are not exactly clear from the figure. For example, what does it mean by the cook->cook task in the sequence level?

**Questions:**

1) What is the difficulty
2) The authors introduced 3 levels of preferences, including action, option and sequence. In section 3.2 it is also mentioned that there are 75 preference tasks from the action level. Why are there no experiments for the action level preferences in the paper?
3) What are these strings (eg. “u7cbw”) in the frame-level text annotation?

---

> ### Author Response · Authors · 2024-11-21
>
> Dear Reviewer rU8v,
>
> We sincerely appreciate your insightful and constructive feedback on our paper.
>
> > 1. Given that the symbol based LLMs work so well compared to the vision based methods, it undermines the utility of the benchmark for preference learning from visual inputs. The example task in Table A.3  gives you an idea about this. I was able to understand the preference of  the user for that task without having access to the video.
>
> Vision observations are utilized to align with real-world scenarios, where textual descriptions of action sequences are rarely available, and vision is the most common method for information acquisition. Using text to feed models is indeed a simplification of the task to minimize difficulties at the perception level and concentrate more on the reasoning and planning aspects of the task. Also, reasoning and planning based on visual inputs is harder. The importance of visual inputs lies in their richness and the depth of context they can provide, which is often not available through textual data alone. Visual cues can offer nuanced insights into user preferences that are not explicitly stated, making them invaluable for applications where understanding subtle, non-verbal user behaviors is critical.
> For the symbol based experiments, as you pointed out, the effectiveness of symbol-based LLMs in preference learning tasks, as demonstrated in our experiments, underscores the relative ease with which these models can handle few-shot induction when provided with explicit action and preference labels. This is acutally a significant finding in the paper as it confirms the potential of LLMs in personalizing user interactions based on textual or symbolic inputs.
>
> > 2. Some of assumptions made at the task level do not make sense to me:   a) How would the agent get access to egocentric observations of the  user? Ideally the robot should learn to pick up preferences by watching a  human perform the task from its own camera.    b) What embodiment is used during the preference demonstrations?   c) Similarly having access to the action sequence of the user.   It would be good to see a discussion around these questions in the  paper.
>
> a) Collecting both egocentric observations and third-person views is feasible in PbP or similar environments built on simulators like iGibson. However, in real-world scenarios, it is generally easier to gather egocentric observations of human daily activities, as these can be efficiently captured through wearable devices. Additionally, there are numerous egocentric-view datasets available, such as Ego4D, which further facilitate this approach. While third-person views can provide a different perspective, they often encounter issues such as occlusion. Although research based on third-person views is essential for applications involving real robots, focusing on egocentric views in the current work allows for a more straightforward exploration of preference learning and planning. Nevertheless, third-person view data can be obtained by integrating additional cameras, as outlined in our provided code.
> b) We employ the Fetch robot for our experiments. The simulator OmniGibson we use does not provide a humanoid robot. And we also think it is unnecessary to adopt a humanoid robot because in our setting, the preference is learnt from sequences of actions like move, pick, place, open, toggle on/off, which is samely displayed whether in a humanoid robot or a fetch robot with an arm.
> c) In experiments involving vision input, we do not explicitly provide the action sequence of the user. In symbolic-based experiment, we provide the action sequence to reduce the perception cost to concentrate more effectively on the inference and planning aspects of the study.
> We have added discussion in Appendix sec B.
>
> > 3. The decision of using Levenshtein distance as the main metric for  the evaluation is arbitrary and penalises the agent for not exactly  matching the training data format. Given that the authors have access to a simulator, I would have liked seeing the evaluation of the actions in the simulator.
>
> It's a very valuable suggestion. We have considered this point in the evaluation process. However, it is unnecessary to evaluate the actions in the simulator as higher-level actions outputted by the model can be executed by the rule-based controllers or pre-defined policies. Therefore, from our view, it's better to directly compare the model-output sequence with the ground truth.

---

> ### Author Response · Authors · 2024-11-21
>
> > 4. Figure 2 can be improved:  a) The text formatting is poor. b) The example tasks are not exactly clear from the figure. For example, what does it mean by the cook->cook task in the sequence level?
>
> Thank you for your suggestion. We will improve the figure in further revision. For b), the sequence level denotes the preference over sub-task order or prioritization of certain sub-tasks over others in one task. "Cook->cook" means that the agent prefers doing cooking tasks continuously.
>
> > Q1. What is the difficulty
>
> The challenge primarily lies in accurately abstracting implicit preferences from minimal observations so that these preferences can generalize across various tasks and effectively guide the planning process. There is a lack of large-scale, well-labeled data for personalized preferences, which necessitates that models learn in a few-shot manner. Also, preferences can be diverse and exist at different levels. Abstracting such preferences involves more than merely computing the distributions of actions, which is just one potential structure. It requires uncovering the hidden abstract structures among minimal observations.
>
> > Q2. The authors introduced 3 levels of preferences, including action,  option and sequence. In section 3.2 it is also mentioned that there are  75 preference tasks from the action level. Why are there no experiments  for the action level preferences in the paper?
>
> In the given context, action preferences are related to specific actions, such as placing an apple in a certain part of a shelf. These preferences are simple to perceive and copy, which means that a basic imitation policy can effectively solve the problem. We would like to focus on more complex preferences that cannot be easily addressed through simple copy/imitation, and to test models in scenarios where preferences involve more nuanced or context-dependent decisions.
>
> > Q3. What are these strings (eg. “u7cbw”) in the frame-level text annotation?
>
> It's the object ID in Omniverse so that the name refers to a unique object in the object, so that the model exactly knows which object we are referring to. For example, there could be more than one apple in the scene.

---

> ### Author Response · Authors · 2024-11-30
>
> Dear Reviewer, thank you again for your time and thoughtful review. Your valuable feedback has been instrumental in helping us refine and enhance our paper.
>
> As the discussion period comes to a close, we would be more than happy to provide additional explanations if you have any concerns or require further clarification.

---

> > ### Comment · Reviewer_rU8v · 2024-12-02
> >
> > Reading the response, I choose to retain my score. I am not fully satisfied with the answers from the authors. The fact that symbol based methods perform so well on the benchmark while only using action description shows the lack of depth of this benchmark. While I agree with the authors that video can provide a lot of rich signal for preferences, the tasks setting in their particular benchmark does not provide this complexity. The preferences that have to be inferred from the video are so very high level that they could have been inferred from the action descriptions. What actually presents a meaning challenge in video based preference learning would be situations where the relevant variables to reason over are tricky to annotate in symbols.

---

### Official Review · Reviewer_aNeE · 2024-11-07

**Soundness:** 2
**Presentation:** 2
**Contribution:** 2
**Rating:** 3
**Confidence:** 3

**Summary:**

The paper introduces the challenge of developing intelligent embodied agents capable of integrating into daily human life by emphasizing the importance of learning individual user preferences. The authors present PbP, an embodied environment built upon Omniverse and OmniGibson simulation platforms, supporting realistic simulation to facilitate the study of personalized preferences and planning. The experiments reveal that existing algorithms struggle to effectively generalize personalized plans from few-shot learning scenarios. The authors propose this work to lay a foundation for further exploration into more efficient and adaptive preference-based planning systems for dynamic environments.

**Strengths:**

1. This paper proposes a new dataset and a simulation environment for the learning and evaluation of embodied agents focused on personalization in planning.
1. The emphasis on human preference and personalization addresses a significant topic in the field of human-centered AI.
1. The experimental findings imply significant challenges for existing algorithms when applied to PbP, underscoring the importance of further research in this area.

**Weaknesses:**

1. The dataset lacks sufficient details and explanation. The definitions and construction of the three levels of preferences are confusing. As demonstrated in Figure 2 and based on my understanding, the preferences are defined as different hierarchical levels of action space, but it is hard to justify how these action spaces relate to human preference or personalization in planning. Without this connection, the dataset appears similar to other few-shot LLM planning datasets. I suggest that the authors include concrete examples of preference-learning tasks and examples for each preference level to better illustrate the dataset's relevance to personalization.

1. According to the prompt format in Appendix D, PbP appears to be a summarization task in which the model classifies a video based on the preference list. Since planning inherently involves predicting future actions, summarizing what has occurred in a video cannot be equated with planning. Please correct me if I have misunderstood this aspect.

1. There is a lack of comparison with state-of-the-art planning methods. The baselines reported are not specialized for planning, and I question whether the prompts have been well-designed for a planning task.

1. The title is somewhat misleading, as it suggests that the paper proposes a new approach for planning. However, it primarily introduces a dataset/benchmark without presenting new algorithms.

1. Other issues: for the reader's convenience, please combine the appendix with the main text into a single PDF file.

**Questions:**

1. What do $f$, $p$, and $g$ represent in Eq 1 and 2 according to the experiments?
1. Is $l$ a loss function in Eq 2?
1. How is the representation $p$ inputted into black-box LLMs like GPT-4(V)?
1. How many demonstrations are used for few-shot learning?
1. Line 255 mentions that there are 15,000 preference-learning tasks. How many videos are included in the dataset in total? How many different personalized planning examples are there for each task?
1. What metric is used in Table 4?

---

> ### Author Response · Authors · 2024-11-21
>
> Reviewer aNeE,
> We sincerely appreciate your insightful and constructive feedback on our paper.
> > 1. The dataset lacks sufficient details and explanation. The  definitions and construction of the three levels of preferences are  confusing ... It is hard to justify how these action spaces relate to human  preference or personalization in planning. I suggest that the authors include concrete examples of  preference-learning tasks and examples for each preference level to  better illustrate the dataset's relevance to personalization.
>
> Thank you very much for your suggestion.
> For the dataset details and explanation, due to the page limit, we have added more information in Appendix Sec.B in the revised paper. We also kindly refer you to the code scripts to see more details about the scene construction and the dataset sampling process.
> For the preference, It's indeed an important question. We are not going cover all human personalized preferences in this work. Actually, preferences defined in our work can be seen as high-level abstractions of action sequences. The generalization of our preference modeling doesn't necessarily depend on the scope of these pre-defined preference definitions. The model learns from action sequences and finally outputs actions as well. The pre-defined preferences mainly serve as a guide to help us sample demonstrations and help model planning. While they may not cover all corner cases, they provide a substantial enough range to serve as a benchmark to evaluate the baselines. Besides, so long as the task can be formulated as action sequences and sub-structures exist in the action distributions, the methodology can be applied, no matter how domains vary, or humans have their own unique preference. For the preference hierarchy, our proposed three-tiered hierarchical structure of preferences is designed from the perspective of how things happen in a household scenario.  As for how these action spaces relate to personalized human preference, we have included a concrete example for each level in sec 3.1 for easier understanding.
>
> > 2. According to the prompt format in Appendix D, PbP appears to be a  summarization task in which the model classifies a video based on the  preference list. Since planning inherently involves predicting future  actions, summarizing what has occurred in a video cannot be equated with  planning. Please correct me if I have misunderstood this aspect.
>
> No. It's a misunderstanding. The paper indeed includes a two-stage progress which can be divided into two distinct tasks: a prediction task and a planning task. The prediction task involves inferring preferences from demonstrations, which might be akin to summarizing past events in a video. However, the planning task, exactly as you noted, involves generating future action sequences based on the observed demonstrations and a new scene. The confusion might have arisen from the fact that only the prompts for the prediction task were shown in Appendix sec D, which might have given the impression that the paper solely focuses on summarization. We have include examples of prompts for both the prediction and planning tasks to clearly differentiate between the two and to illustrate how the planning task involves generating plans for future actions in revision.
>
> > There is a lack of comparison with state-of-the-art planning  methods. The baselines reported are not specialized for planning, and I  question whether the prompts have been well-designed for a planning.
>
> Our focus in this paper is primarily on evaluating the planning capabilities of large language models (also MLLMs). These models are particularly adept at leveraging common sense and prior knowledge, which are crucial for effective planning in dynamic and nuanced scenarios. This inherent capability makes them suitable candidates for preference-related tasks we are exploring. Regarding the trend in the research community, there is growing interest and promising results in employing LLMs for few-shot and zero-shot planning tasks. This approach is supported by recent studies such as [1, 2, 3], which demonstrate the potential of LLMs in these areas. As for the prompts used in our experiments, we have adopted them from OpenAI’s official prompt cookbook.
> [1] Song, C. H., Wu, J., Washington, C., Sadler, B. M., Chao, W.-L., and Su, Y. (2023). Llm-planner: Few-
> shot grounded planning for embodied agents with large language models. ICCV.
> [2] Driess, D., Xia, F., Sajjadi, M. S., Lynch, C., Chowdhery, A., Ichter, B., Wahid, A., Tompson, J., Vuong, Q.,
> Yu, T., et al. (2023). Palm-e: An embodied multimodal language model. ICML.
> [3] Yao, S., Zhao, J., Yu, D., Du, N., Shafran, I., Narasimhan, K., & Cao, Y. (2023, January). ICLR.

---

> ### Author Response · Authors · 2024-11-21
>
> > 4. The title is somewhat misleading, as it suggests that the paper  proposes a new approach for planning. However, it primarily introduces a  dataset/benchmark without presenting new algorithms.
>
> We are sorry for the confusion. What we want to stress in the title is that preference can serve as a guidance for planning. To prove this, we propose a benchmark to incorporate several preference-based-planning tasks, and test baseline models on it. For the above considerations, we choose the Primary Area "datasets and benchmarks." We will definitly consider modifying the title if necessary.
>
> > 5. Other issues: for the reader's convenience, please combine the appendix with the main text into a single PDF file.
>
> Thank you very much for the practical advice. We have combined the appendix in revised version.
>
> > Q1. What do f,p, and g represent in Eq1 and 2 according to the experiments?
>
> In Eq 1, $f$ denotes the function used to predict the preference $p$ from the observation. $\theta_f$denotes its parameters. $p$ denotes the preference representation, see l298. $g$denotes the planning function, see l305. In experiments, $f$and $g$denote the stage 1 and 2 model in sec 5.3, and  $p$ denotes the preference labels.
>
> > Q2. Is l a loss function in Eq2?
>
> Yes. We have added notations.
>
> > Q3. How is the representation p inputted into black-box LLMs like GPT-4(V)?
>
> For LLMs, they are explicit textual labels.
>
> > Q4. How many demonstrations are used for few-shot learning?
>
> 3 demonstrations are used for few-shot learning.
>
> > Q5. Line 255 mentions that there are 15,000 preference-learning tasks.  How many videos are included in the dataset in total? How many different  personalized planning examples are there for each task?
>
> We are sorry for the confusion. Line 255 actually means that there are 15,000 unique preference-learning instances (videos), as each of them is randomly sampled in the simulator under one preference. See Appendix sec B.1 for the statistics. We have modified the statement.
>
> > Q6. What metric is used in Table 4?
>
> The accuracy of preference prediction, as shown in Table 2 and 3. We have added further notations in revision.

---

> ### Author Response · Authors · 2024-11-30
>
> Dear Reviewer, thank you again for your time and thoughtful review. Your valuable feedback has been instrumental in helping us refine and enhance our paper.
>
> As the discussion period comes to a close, we would be more than happy to provide additional explanations if you have any concerns or require further clarification.

---

### Author Response · Authors · 2024-11-22
**Overall Response**

We are deeply appreciative and grateful for the time and effort the ACs and reviewers have invested in evaluating our work and providing valuable, thoughtful, and constructive feedback. We are particularly thankful to the reviewers for recognizing the high novelty of our contribution (Reviewer nfLL), acknowledging that we address a significant issue in the field of human-centered AI (Reviewer aNeE), and noting our proposal of a well-designed and useful benchmark (Reviewer SCvY). In the following sections, we have endeavored to address the specific questions from each reviewer to the best of our ability. We have also revised our paper based on the suggestions provided. We kindly refer the reviewers to the sections we have noted for their review.

---

### Meta-Review · Area_Chair_1xrW · 2024-12-19

**Metareview:**

While PbP presents an interesting attempt at creating a benchmark for personalized preference learning in embodied AI, several fundamental issues remain insufficiently addressed. The most critical concern is that the benchmark's utility is undermined by symbol-based methods significantly outperforming vision-based approaches, suggesting the visual components may be unnecessary. Additionally, the pre-defined preference vocabulary and rule-based generation of few-shot examples raise questions about real-world applicability. While the authors attempted to address these concerns, their responses about egocentric observations and justification for visual inputs remain unconvincing, particularly given the superior performance of text-only baselines. I recommend rejecting this paper.

**Additional Comments On Reviewer Discussion:**

During the rebuttal period, the reviewers raised concerns about PbP's questionable utility of visual components, ambiguity in preference definitions, and insufficient ablations. While the authors provided additional studies and argued for visual inputs' importance, their responses remained unconvincing. Some reviewers maintained low scores, emphasizing that visual components add unnecessary complexity given text-based methods' superior performance. Though others were more positive, the fundamental issues about the benchmark's ability to capture real-world preference learning and the strong performance of simple text-based approaches weren't adequately addressed, suggesting substantial revision is needed before publication.

---

### Decision · Program_Chairs · 2025-01-22

Reject